# Uphill Flow Rock Ramps. How the Design Impacts Their Functionality

**Leticia Carrero-Díez \*, Carolina Martínez Santa-María and J. Anastasio Fernández-Yuste**

Department of Forest and Environmental Engineering and Management, School of Forest Engineering and Natural Resources, Universidad Politécnica de Madrid, C/José Antonio Novais 10, 28040 Madrid, Spain
\* Correspondence: leticia.carrero@upm.es

**Abstract:** Enhancing river passability is considered a central part of the efforts to maintain fish population and achieving good ecological status, according to the EU Water Framework Directive (WFD). One commonly proposed approach to achieving this aim involves the creation of fishways. However, recent studies have shown that many of these fish passes are often lacking an optimal design with far reaching consequences for fish migration. Several promising new designs such as nature-like rock ramps, with uphill flow, have been recently developed. Such studies attempt to address these drawbacks by adapting several structure-related features (i.e., boulder size and shape and friction walls). In this study, we used a 2D computational fluid dynamic model to assess how the key hydraulic variables (water depth, velocity and turbulent kinetic energy) were impacted by different design elements of uphill flow fishways with different configurations. With regard to the standard boulder shapes and sizes, our results reveal that: (1) doubling the boulder sizes results in a decrease of turbulent kinetic energy in resting corridors of up to 33%; (2) the inclusion of small friction-walls in the ramp design increases uphill velocity in the intermediate corridors by up to 49%; (3) the trapezoidal shape of the boulder leads to the largest decreases in maximum velocity in the gaps (16%) and the largest increases in the uphill velocity in the resting corridors (180%). These results may allow us to optimise the uphill flow rock-ramp design to improve the passability of this type of fishways.

**Keywords:** uphill flow; passability; boulder placement; wall-friction; flow velocity; 2D model

## 1. Introduction

Many of the aquatic organisms in rivers migrate in order to complete their biological cycle. These migrations allow them to access habitats with suitable environmental conditions (i.e., water temperature, dissolved oxygen) and suitable hydraulic biotopes (depth, velocity and roughness) to take advantage of both high-quality breeding sites and bountiful feeding areas [1]. The fragmentation of rivers through barriers on lotic systems has had negative consequences on riverine ecosystems. Therefore, the viability of many fluvial species depends on the functional connectivity of their populations through dispersal. Transversal barriers induce localized changes that alter the continuum of stream temperature, water chemistry, energy, sediment and aquatic organism movements [2,3]. One commonly proposed approach to conserve connectivity or to mitigate the fragmentation is the construction of fish passages. There are a number of diverse topologies of fish passages [4–6]. Despite the clear importance of their use, their optimal design remains an object of intense debate and research [7]. Therefore, there are a myriad of studies in the literature focused on assessing aspects such as fishway attraction and entrance, and passage efficiency [8]. Nature-like fishways are currently very popular across the world as an alternative passage system with comparable efficiencies to technical fishways [9].

Rock ramps, or more commonly, ramps, are one of the most highly recommended passage devices for low-height obstacles due to their naturalized nature, their versatility,

and their high degree of passability by fish fauna [10]. They consist of a long gentle sloping channel with a naturalized substrate and interspersed boulders, designed to maintain biologically suitable depth and velocity conditions over a limited range of water level changes. Their hydraulic conditions are dependent on the slope of the channel and by the arrangement and size of the boulders. Regarding the arrangements of the boulders, two approaches are widely used: longitudinally aligned in rows perpendicular to the main flow; or randomly distributed [11]. In these ramps, the water flows through the spillways, or the existing notches/gaps, between two consecutive boulders. The critical conditions for fish passability are given by the velocity in the flow gap, the minimum depth achieved in the channel, the jump in the cases of no submerged passage, and the power generated in the water drop and its dissipation to fulfil suitable thresholds. The design of these ramps must ensure certain hydraulic conditions of the structure for the operating flows. The evaluation of the suitability of these hydraulic conditions must be conducted through the variables mentioned above, ensuring that their ranges of variation are compatible with the swimming capacities of the target species [4,11–15]. Moreover, recent studies including both fish behaviour and hydraulics in detail, and have pointed out the importance of other, more specific, hydraulic variables, such as velocity distribution, turbulent kinetic energy, turbulent intensity and vorticity to improve the design, evaluation and efficiency of these type of ramps [16–24].

A new design for rock-ramps has recently been developed: uphill flow rock-ramps [25]. The uphill flow ramps, while keeping many of the elements of the traditional rock-ramps, are able to generate a very particular hydraulic performance, as they induce uphill flows paths in opposition to the main flow in the ramp. This is achieved through their particular zig-zag distribution of boulders within the row (Figure 1).

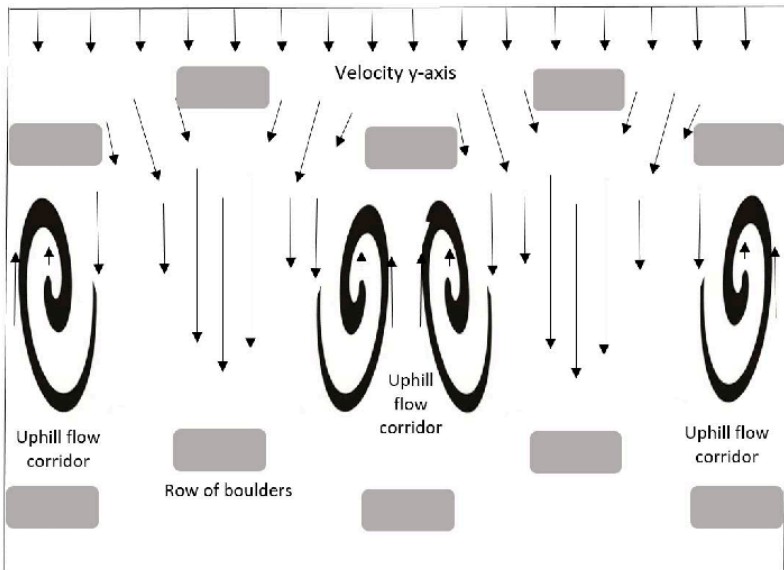

**Figure 1.** Illustration of resulting uphill flows due to zig-zag boulder distribution.

The biological importance of these new uphill flow paths shows great potential as they favour the progression of upstream fishes to such an extent that they could advance up the ramp, pushed by these uphill flows, and would only have to swim to pass the gap between two boulders.

An uphill flow ramp design has already been built in the Manzanares river (Madrid, Spain), but neither hydraulic variables data nor fish passage records are currently available. However, new uphill flow ramps are currently under planning in diverse rivers belonging to the Duero, Guadalquivir, Guadiana and Miño-Sil river basins (Spain). Updated one-dimensional equations [26] considering non-uniform flow analysis [27] have been used for their design [25].

The main objective of this research was to identify alternative designs that improve the functionality of uphill flow rock ramps by applying IBER, an bidimensional model [28]. In this paper, four different uphill flow ramp configurations were performed. We aimed to answer three questions: Do changes in boulder dimensions improve uphill flow paths conditions and reduce the turbulent kinetic energy? How can the incorporation of friction-increasing walls improve the uphill velocity conditions and decrease the turbulent kinetic energy? Does the shape of the boulder reduce the maximum water velocity in the flow gaps? The results of this study may be useful to fish pass planners for enhancing the design of uphill flow rock-ramps, concerning their functionality and, hence, improving their passage efficiency.

## 2. Materials and Methods

### 2.1. Description of the Geometrical Designs

The present paper studied an uphill flow ramp, designed by upflow ramps dimensioning assistant [29]. We defined four different ramp layouts in order to numerically investigate the previously formulated hypothesis. The first layout (R3) reproduced the hydraulic behaviour of a standard three module uphill flow ramp (Figures 2 and 3). The other three layouts replicated the standard ramp with some geometric modifications, in order to aim at exploring the effect of: (1) boulder dimensions (Layout DR3); (2) friction (Layout DRb3); (3) shape of boulders (Layout DT3). These layouts (Figures 4–6) allowed direct comparison between the standard ramp (R3) and the other designs, and allowed us to verify, using a numerical model performance, their effects over some specific hydraulic variables.

The generation of the uphill flow is guaranteed, as long as the boulders in the rock-row are distributed with a certain disposition, as exposed in Figure 1. This distribution causes the flow through the pools to be compartmentalized. Then areas with uphill flow are interspersed with downhill flow areas, which are referred to as corridors. The uphill flow corridors are always located in the vicinity of the walls due to the influence of the roughness of the contour, and are under the protection of the down face of the lowest positioned boulder in each row.

With this boulder distribution in mind, the four different layouts were designed.

- Rectangular boulders (R3)

This study used a standard ramp (R3) with rectangular boulders and three modules. A module is the minimum unit required to generate uphill flow and is formed by three consecutive boulders, belonging to one particular rock-row and placed in the correct distribution. The uphill flow ramp design comprises three phases: hydraulic dimensioning, geometric dimensioning, and power dissipation dimensioning. The most relevant data for the hydraulic dimensioning is the minimum flow for which the ramp must be functional. In this case, 2 $m^3$ $s^{-1}$ was fixed by considering the construction of three modules for the design and ensuring a minimum depth. The dimensioning was conducted in a uniform regime, assuming that the depths upstream and downstream of the boulders are the same in all rock-rows, being 0.6 and 0.3 m, respectively, and with a geometric gap width (WGg) of 0.35 m. Using the aforementioned values, the geometric data were generated by the application, fixing a height difference on the ramp bed between the inlet and outfall of 1.9 m in seven rows and six pools. Considering a compatible boulder diameter (Db) of 0.7 m, a boulder width (Wb) of 0.5 m, and the angle ($\alpha$) defining the alignment between two consecutive boulders of 45 degrees and fixing 0.06 as the slope, the values of pool length and width ($L_P$ = 5 m $W_P$ = 6.4 m) were generated as well as the pool volume (5 $m^3$) and the dissipated power (400 W $m^{-3}$). Finally, the model domain was 55 m long with six target pools, one outfall pool (with double length) and eight rock-rows (the last one is the control row). The control row and the outfall pool are used to guarantee a steady flow. Due to the influences that the inlet and outlet conditions could have in the target pools, extra upstream and downstream pools were added in the ramp layout. A representation of the R3 layout domain can be seen in Figure 2 and the details of a pool and a rock-row dimensions are shown in Figure 3.

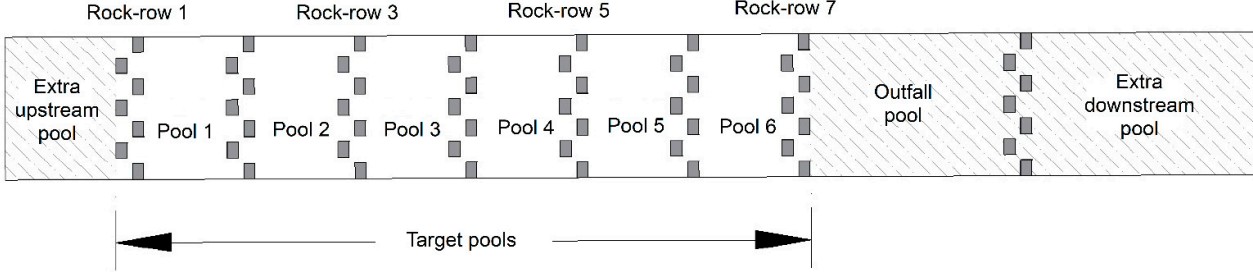

**Figure 2.** Rectangular boulder layout (R3) for 2D model domain from a top view.

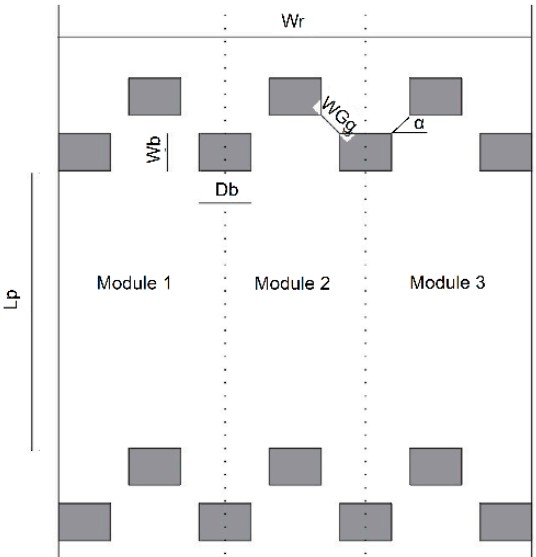

**Figure 3.** Details of a pool and a rock-row geometry of rectangular boulder layout (R3).

- Double Rectangular boulders (DR3)

The wider the boulder, the more uphill flow surface area it will generate. To this end, a R3 case was reproduced, but the boulder diameter of the downiest positioned boulder in each row was duplicated (Db = 1.4 m). As result, from the seven corridors, the odd ones were wider ($W_c$), and these were the corridors where the uphill flows occurred (Figure 4).

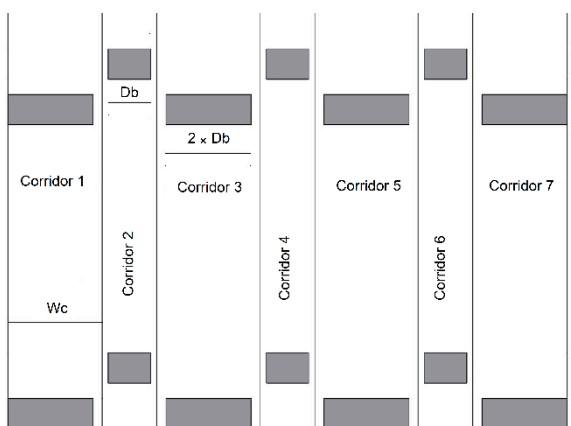

**Figure 4.** Schematic diagram of boulders and corridors geometry of double rectangular boulder layout (DR3).

- Double Rectangular boulders with a small barrier (DRb3)

The decrease in velocity caused by friction is widely known. This layout was designed to assess the effect of friction by incorporating small friction walls in the mid-boulders (corridors 3 and 5) of the DR3 arrangement, as show Figure 5. These barriers were 0.2 m width and 2 m long.

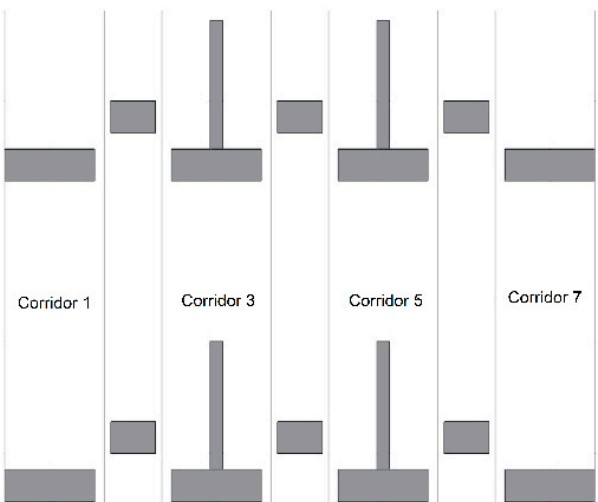

**Figure 5.** Details of friction walls position of double rectangular boulder with a small barrier layout (DRb3).

- Double Trapezoidal boulders (DT3)

This layout was configured in order to evaluate the effect of the boulder shape on the maximum velocity in the gaps. Among the works found in the literature, spherical and square-shaped obstacles were used [16,17,30,31]. The aim here was to investigate how the trapezoidal shape impacted the uphill flow areas as it maximizes the diameter downhill of the boulder, and then of the uphill flow areas. For this purpose, new boulders were designed with 1.4 m of diameter (Db) and 1.2 m of the shorter diameter (Db*) for the wider boulders, while the first row of each rock-row were 0.7 and 0.5 m, respectively (Figure 6).

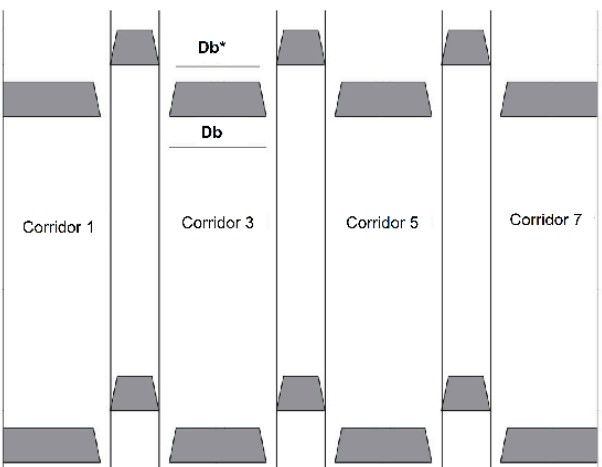

**Figure 6.** Detail of trapezoidal boulder dimension of double trapezoidal boulder layout (DT3). * indicates the shorter base of the trapezoid.

The different design parameters are listed in Table 1. A total of four configurations were conducted.

**Table 1.** Layout parameters of every comparison scheme: rectangular boulder (R3); double rectangular boulder (DR3); double rectangular boulder with small barriers (DRb3) and double trapezoidal boulder (DT3). * The number indicate the corridor position.

| Layout | Boulder Diameter (m) | Shorter Boulder Diameter (m) | Boulder Width (m) | Geometric Gap Width (m) | Ramp Width (m) | Corridor Width 1, 7 * (m) | Corridor Width 2, 4, 6 * (m) | Corridor Width 3, 5 * (m) |
|--------|--------|--------|--------|--------|--------|--------|--------|--------|
| R3 | 0.7 | 0.7 | 0.5 | 0.35 | 6.4 | 0.825 | 0.95 | 0.95 |
| DR3 | 0.7/1.4 | 0.7/1.4 | 0.5 | 0.35 | 9.2 | 1.525 | 0.95 | 1.65 |
| DRb3 | 0.7/1.4 | 0.7/1.4 | 0.5 | 0.35 | 9.2 | 1.525 | 0.95 | 1.65 |
| DT3 | 0.7/1.4 | 0.5/1.2 | 0.5 | 0.35 | 8.575 | 1.5 | 0.7 | 1.7 |

*2.2. Numerical Model*

The flow pattern in the pools is of great importance for guiding the fish through the fishway. Such flow generates complications because they create separation flow effects around obstacles with wake interactions. Therefore, the 3D flows appear to be dominant and they have been investigated by many authors [32–34]. However, an efficient 3D model necessitates using large adaptive meshes, making CPU time consuming. The flow field in several designs of vertical slot fishways (VSF) and rock-weir fishways were also studied experimentally and numerically with 2D models by Puertas et al. [35], Marriner [36] and Tran [22], among others. By comparing the numerical and experimental results, Cea [37] exposed that the depth-averaged shallow water equations, with a suitable turbulence model, could be used in order to calculate these variables in vertical slot fishways (VSF). Cea included in its analysis RANS 3D depth-averaged with k-ε roughness model. Fuentes-Perez [37] demonstrated that RANS 3D with k-ε slightly underestimated TKE with respect to LES (large eddy simulation method using the Smagorinsky turbulence model), but advocated the use of RANS to simulate larger spatial scales corresponding to the time-averaged flow, and LES in regions where a more detailed analysis was required. In our study, we considered that the flow behaviour in VSF and in uphill flow rock ramps presented similar hydraulic characteristics: (i) the inflow into the pool went through flow gaps without bottom orifices and without weirs; (ii) the gap was narrow in comparison to the width of the pool; (iii) and the energy dissipation in the pool was linked to the turbulent kinetic energy. These similarities allowed us to assume that the results of Cea and Fuentes are also valid for the type of fishway studied in this work. Tran [22] also opened up the possibility of using a depth-averaged shallow water model in nature-like fishways in order to investigate design options. This study could show that a 2D depth integrated model can help the designer when the rock ramp has less than 7% slope and a boulder concentration between 6–20%. Consequently, the 2D model, which is quite easy to use, appears to be an appropriate tool for considering special geometrical configuration and for adapting designs, and could lead to a more detailed examination (3D modelling) of a final design for nature-like fishways.

In this study, we used IBER 2.5.2 [28] (depth-averaged model with k-ε model turbulence) to obtain depths, velocity and TKE fields. It has been successfully used for a wide range of applications, including the modelling of river flow and quality [38], instream wood transport [39], geomorphic impacts of dam failure [40] or performance vertical slot fishways [41]. The model solves the depth-averaged shallow water equations coupled with a turbulence model. These equations were obtained after vertical integration of the three-dimensional Reynolds averaged Navier-Stokes (3D-RANS) equations over the water depth (see further details in Blade et al. [28]). For turbulence modelling, this work employed k-ε model from Rastogi and Rodi [42]. These equations are given by

$$\frac{\partial h}{\partial t} + \frac{\partial hU_x}{\partial x} + \frac{\partial hU_y}{\partial x} = Ms$$

$$\frac{\partial hU_x}{\partial t} + \frac{\partial hU_x^2}{\partial x} + \frac{\partial hU_xU_y}{\partial y} = -gh\frac{\partial Z_x}{\partial x} + \frac{\tau_{s,x}}{\rho} - \frac{\tau_{b,x}}{\rho} - \frac{g}{\rho}\frac{h^2}{2}\frac{\partial p}{\partial x} + 2\Omega\sin\lambda U_y + \frac{\partial h\tau_{xx}^e}{\partial x} + \frac{\partial h\tau_{xy}^e}{\partial y} + M_x$$

$$\frac{\partial hU_y}{\partial t} + \frac{\partial hU_xU_y}{\partial x} + \frac{\partial hU_y^2}{\partial y} = -gh\frac{\partial Z_s}{\partial y} + \frac{\tau_{s,y}}{\rho} - \frac{\tau_{b,y}}{\rho} - \frac{g}{\rho}\frac{h^2}{2}\frac{\partial p}{\partial y} - 2\Omega\sin\lambda U_x + \frac{\partial h\tau_{xy}^e}{\partial x} + \frac{\partial h\tau_{yy}^e}{\partial y} + M_y$$

where h is depth; $U_x$ and $U_y$ are the depth averaged horizontal velocities; g is the gravity acceleration; Z is the free layer elevation; $\tau s$ is the free surface friction due to wind induced friction; $\tau b$ is the bed friction; $\rho$ is the water density; $\Omega$ is the earth's rotation angular velocity; $\lambda$ is the latitude of the studied point; $\tau_{xx}^e$, $\tau_{xy}^e$, and $\tau_{yy}^e$ are the effective horizontal tangential stresses; and Ms, $M_x$, and $M_y$, respectively, are the terms of mass source/drain and momentum, which are used to model precipitation, infiltration and drainage.

Boundary conditions were applied to the inlet (discharge Q = 2 m³ s⁻¹) and the outlet in critical/subcritical conditions, while the wall condition was employed for the bottom boulders and side walls (Manning coefficient equal to 0.02 s m⁻¹ᐟ³). This coefficient is recommended by Zheng [10] for roughness particle sizes of 2 cm, which is the worst case scenario (lower roughness values implies higher velocities). The four ramp layouts were transformed into a regular calculation mesh, with a resolution equal to 0.05 m using the Geospatial Data Abstraction Library (GDAL) approach. The simulations were stopped as soon as the hydrodynamics reached the steady condition (the difference between the inlet and outlet flows is negligible).

### 2.3. Hydraulic Variables Monitoring

Changes in hydraulic conditions, as reflected in water velocity, turbulence characteristics, and momentum, cause the mayor cue fish to seek a migration pathway in rivers [43]. However, these variables need to be considered in order to design fishways that are easily passable.

The design and the dimensioning of these rock-ramp models were planned to ensure uphill flow areas. However, the different layouts provided diverse values of the hydraulic variables. The three variables involved in this study were: uphill component of velocity vector (vy); maximum velocity throughout the flow gap (vmax); and the turbulent kinetic energy (TKE).

The uphill component of velocity vector represents the "y" component of the velocity vector, where the "y" axe follows the longitudinal direction of the ramp. Positive values indicate uphill flow and negative values show downhill flow.

Iber estimates the turbulent kinetic energy as TKE = $\frac{1}{2}\left(\left(\overline{u}'\right)^2 + \left(\overline{v}'\right)^2\right)$.

In addition to the control variables listed above, the depth (h) was also collected to ensure that the different model configurations did not result in drastic changes, which would potentially impact on the passability of the ramp. Although the simulations offered the values of these variables in each cell of the calculating mesh, their values were only relevant in certain considered zones for evaluation. Table 2 presents the zones of the ramp where their values were taken for each variable and the justification for this zoning.

Once the data were obtained, the differences were analysed. We assessed the difference between variable data found in the standard ramp (R3) and in the other layouts. Initially, for the R3 ramp analyse, we considered all pools (1–6). Then, we observed that some variables had significant differences, depending on the pool, because they were under the influence of boundary conditions at the end of the simulation. Therefore, we assumed only the stable pools for the analysis.

**Table 2.** Primary details of the zones where the hydraulic variables were taken.

| Hydraulic Variable | Zone | Cell Numbers in R3 | Justification |
|---|---|---|---|
| Depth<br>h<br>(m) | Pool | Pool = 1820<br>$Total_{ramp}$ = 10,920 | Depth was measured in every target pool because the fish must be guaranteed a minimum depth that allows it to move easily and safely. |
| Uphill component of velocity vector<br>vy<br>(m s$^{-1}$) | Corridor | Pool<br>Corridor 1, 7 = 351<br>Corridor 2, 4, 6 = 351<br>Corridor 3, 5 = 390<br>$Total_{ramp}$ = 15,210 | The velocity on the flow direction (y-axis) was measured in each of the seven corridors into which every pool was divided (positive values indicate uphill flow and negative values show downhill flow).<br>As a consequence of the zigzag distribution of the boulders in the row, uphill secondary flows were generated in the corridors 3 and 5 where slow, null or even uphill flow velocities may have been generated.<br>Corridors 1 and 7 were affected not only by the uphill behaviour but also by the roughness of the contour (higher uphill flows and lower TKE values).<br>The flow through the spillways generated corridors (2, 4 and 6) where the downhill velocity and the TKE were high (fast corridor). |
| Maximum velocity<br>vmax<br>(m s$^{-1}$) | Flow gap | Rock-row = 294<br>$Total_{ramp}$ = 2058 | The critical conditions of fish passability were given by the velocity in the flow gap. |
| Turbulent kinetic energy<br>TKE<br>(m$^2$ s$^{-2}$) | Corridor | Pool<br>Corridor 1, 7 = 351<br>Corridor 2, 4, 6 = 351<br>Corridor 3, 5 = 390<br>$Total_{ramp}$ = 15,210 | Same justification as uphill velocity vector. |

Statistical analyses were realized with the software IBM SPSS Statistics Base 25. The significance of differences among the sets (i.e., between pools or layouts) was contrasted through the nonparametric Kruskal-Wallis test, for two or more groups, respectively. When testing differences between more than two groups, we identified which pairs of groups

were different by applying a post hoc pairwise comparison test with the Dunn-Bonferroni correction for multiple testing. Significance was set to *p*-value < 0.05.

## 3. Results

### 3.1. The Influence of Pool, Corridor and Rock-Row Position (R3)

We observed a relationship between the location of pools, the depth (h) and the TKE (Figure 7a,b). The Bonferroni test, with a *p*-value lower than 0.001, showed significant differences between the values found in pool 1 and the other pools. Lower depth and TKE values were reached in pool 1 in comparison the others. The differences in median were 13% and 42% for depth and TKE. The pattern of maximum velocity (Figure 7c) concluded that the values of this variable were independent of their row location.

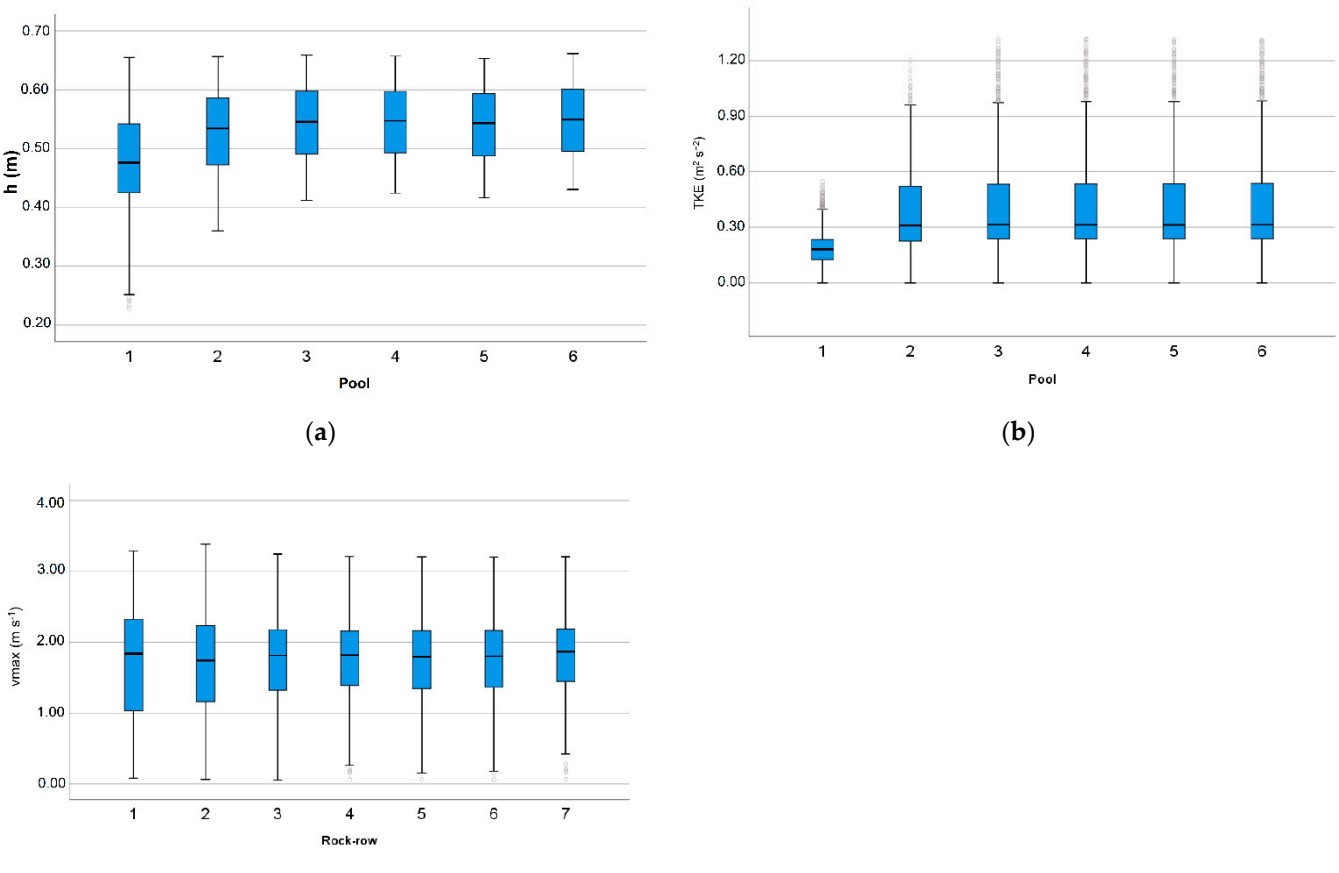

**Figure 7.** Boxplot of ensemble results of numerical simulation for rectangular boulder layout (R3). (**a**) water depth; (**b**) turbulent kinetic energy; and (**c**) maximum velocity in flow gaps. The boxplot represents the minimum, maximum, median, first quartile (25th percentile), third quartile (75th percentile) and outliers.

The test allowed us to rule out pool 1 because the control variables between that pool and the others showed significant differences. The records of this pool were disregarded because the control variables may have been affected by instabilities related to the inflow boundary condition, variabilities that could mask the effect of the design modifications.

The analysis of the corridors (Figure 8) showed a clear difference in the values, according to their location. The uphill velocity revealed differences (*p*-value < 0.001) between corridors 1–7, 2-4-6, and 3–5. According to the values of the variables, it was possible to identify resting corridors (corridors 1 and 7), where velocities were close to 0; fast paths with high downhill velocities (median value of −0.86 m s$^{-1}$); and intermediate corridors,

where the median was $-0.66$ m s$^{-1}$ (corridors 3 and 5). The following table (Table 3) shows the *p*-values when the Kruskal-Wallis test did not allow to reject the null hypothesis.

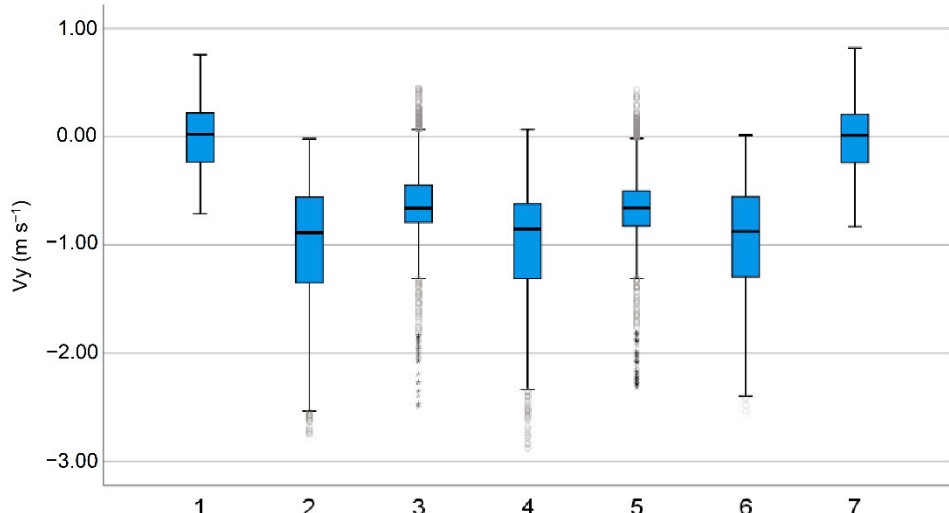

**Figure 8.** Boxplot of uphill velocities by corridors when rectangular boulder layout (R3) was simulated. Positive values indicate uphill flow and negative values show downhill flow.

**Table 3.** *p*-Values from Kruskal-Wallis test when uphill velocities from the different corridors are compared.

|            | Corridor 4 | Corridor 5 | Corridor 6 | Corridor 7 |
|------------|------------|------------|------------|------------|
| Corridor 1 |            |            |            | 0.582      |
| Corridor 2 | 0.238      |            | 0.479      |            |
| Corridor 3 |            | 0.064      |            |            |
| Corridor 4 |            |            | 0.059      |            |

### 3.2. Influence of Designs in Water Depth

The test found significant differences (*p*-value < 0.001) in depth for each simulated design. The median values were: 0.546 m in R3 layout (interquartile range: 0.49–0.59 m); 0.539 m in DR3 (interquartile range: 0.49–0.59 m); 0.532 m and 0.526 m in DRb3 (interquartile range: 0.48–0.58 m); and DT3 (interquartile range: 0.48–0.58 m). Although statistical differences were found, the design effect on depth was not relevant for fish passability.

### 3.3. Influence of Designs in Uphill Velocity

We observed a robust relationship between the uphill velocities and the ramp designs. Figure 9 gives the contours of uphill velocity for three designs: R3 (a), DRb3 (b), and DT3 (c). In this case, three different regions could be distinguished: zones of high downhill velocities colored in red (vy< $-0.5$ m s$^{-1}$); a secondary region (yellow) of low downhill velocities ($-0.5 <$ vy $< 0$ m s$^{-1}$); and a third area (green), with clear uphill velocities (vy $> 0$ m s$^{-1}$). Across the designs, the highest values of uphill velocity are found in corridors 1 and 7, near the contour of the pools. Under the protection of the double boulders, larger areas with uphill velocities are also found (Figure 9b,c) where corridors three and five are identified because of the recirculation pattern effect. In this sense, these areas are even larger when the wall-friction barrier are included in the design (DRb3). In order to compare the size occupied by each type of region, Table 4 was created. The data show twice the uphill velocity flow area in ramps with double boulder than with single boulder.

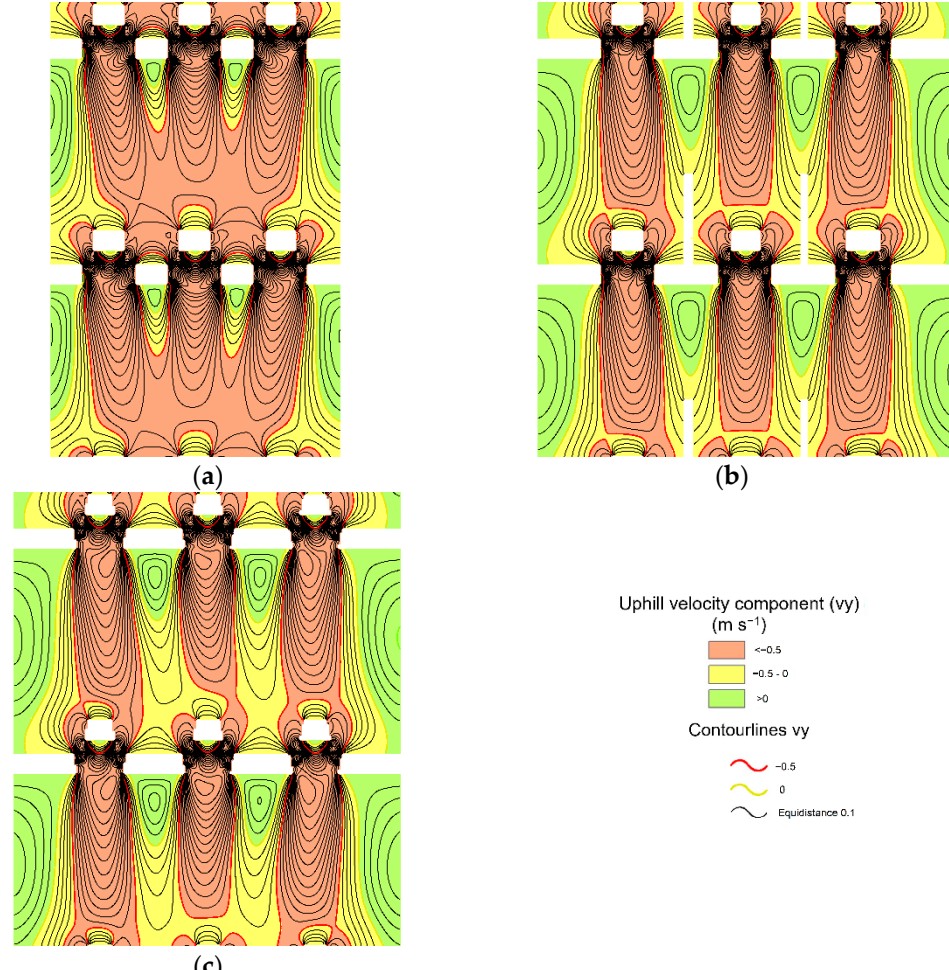

**Figure 9.** Uphill component of velocity vector in pool 3 and 4 for rectangular boulder layout (**a**), double rectangular boulder with a small barrier layout (**b**) and double Trapezoidal boulders (**c**).

**Table 4.** Percentage of flow area occupied by each uphill velocity range and layout type: rectangular boulder layout (R3), double rectangular boulder (DR3), double rectangular boulder with a small barrier layout (DRb3) and double trapezoidal boulders (DT3). The percentage has been calculated respect of the pool area.

| vy (m s$^{-1}$) | R3 | DR3 | DRb3 | DT3 |
|---|---|---|---|---|
| $\leq -0.5$ | 63.05 | 38.69 | 42.51 | 42.94 |
| $-0.5$–$0$ | 25.50 | 40.96 | 34.03 | 33.14 |
| $\geq 0$ | 11.45 | 20.35 | 23.46 | 23.92 |

The numerical results show similar trends, although the pattern was slightly different for fast corridors (Figure 10). The double boulder (DR3) layout presented higher velocities in the y-axe than R3. In the intermediate corridors, vy increased from $-0.65$ to $-0.42$ m s$^{-1}$, where these values are $-0.05$ and $0.02$ in the resting corridors. Overall, the influence of the friction wall improved the functionality of the ramp (DRb3 vs R3). This is particularly relevant in the intermediate corridors, where the uphill velocity module decreased (vy has increased from $-0.65$ to $-0.33$ m s$^{-1}$). In the DT3 design, the median vy reached its highest value, $0.04$ m s$^{-1}$, in the resting corridors. In the intermediate corridors, the shape of the boulders had the same effect as the boulder dimension, decreasing the downhill velocity modules (vy increased from $-0.65$ to $-0.43$ m s$^{-1}$), while in fast corridors, the downhill velocity module was increased (vy decreased from $-0.85$ m s$^{-1}$ in R3 to $-1.08$ in DT3).

**Uphill velocity component vy (ms⁻¹)**

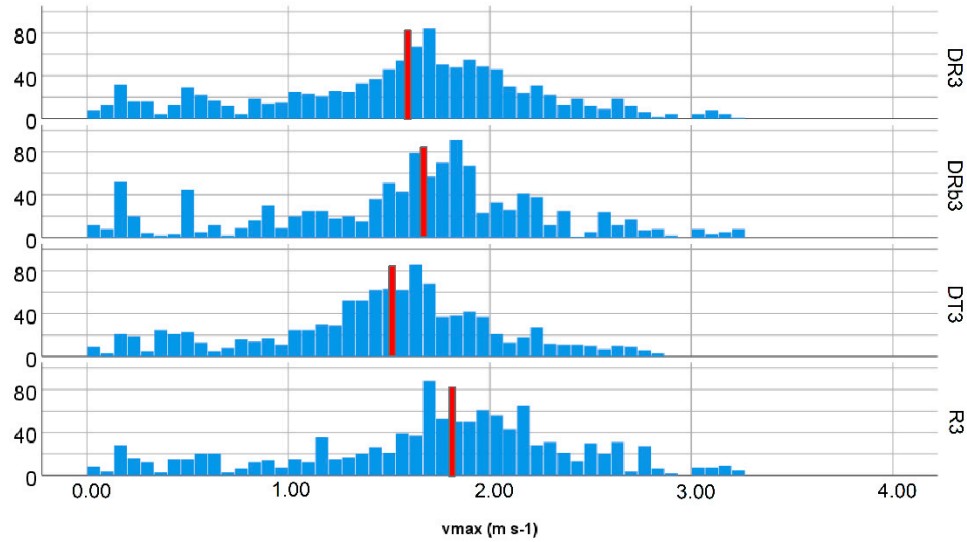

**Figure 10.** Median values of uphill velocity component by corridor type: fast (FAST), resting (REST) and intermediate (INT) and design layout: rectangular boulder (R3), double rectangular boulder (DR3), double rectangular boulder with a small barrier (DRb3) and double trapezoidal boulder (DT3). Positive values indicate uphill flow and negative values show downhill flow.

### 3.4. Influence of Designs in Maximum Velocity

The DR3 simulation generated lower values of maximum velocity in the gaps: 1.65 m s⁻¹, compared to 1.81 in R3 (Figure 11). The maximum velocity differences were also noticeable in the R3 layout, although the barrier had a negative impact because this difference is slightly lower (0.12 m s⁻¹) than with DR3 design. The reduction in maximum velocity in the gaps is a great contribution of the DT3 design. The simulation using a trapezoidal shape for the boulders showed a median value of 1.52 m s⁻¹, decreasing this value 0.29 m s⁻¹ in relation to R3.

**Figure 11.** Histogram of maximum velocity by design layout: rectangular boulder (R3), double rectangular boulder (DR3), double rectangular boulder with a small barrier (DRb3) and double trapezoidal boulder (DT3). The red line represents the median values.

### 3.5. Influence of Designs in Turbulent Kinetic Energy

In relation to TKE, in intermediate corridors the median values in the R3 and DR3 layouts were 0.39 and 0.31 m² s⁻², respectively, and 0.18 and 0.12 m² s⁻² in resting zones. The influence of the friction walls also enhanced the turbulent values in the intermediate corridors, with a reduction of 0.11 m² s⁻² in the R3 design (Figure 12).

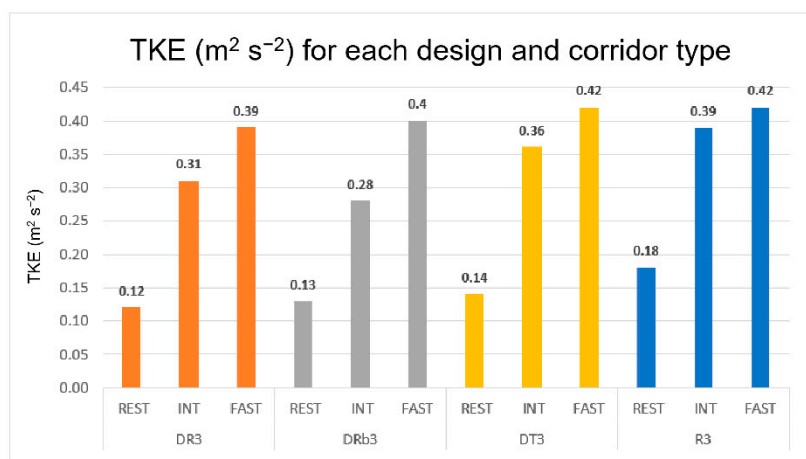

**Figure 12.** Median values of turbulent kinetic energy by corridor type: fast (FAST), resting (REST) and intermediate (INT) and design layout: rectangular boulder (R3), double rectangular boulder (DR3), double rectangular boulder with a small barrier (DRb3) and double trapezoidal boulder (DT3).

Similar to the uphill velocity, the distribution of TKE values also shows a well-defined distribution pattern (Figure 13). It is noteworthy that the double rectangular boulder with a barrier design was able to open lower-energy corridors in the center of the pool. In these intermediate corridors, the TKE decreased under 0.3 m² s⁻² and even areas appeared, under the boulder protection, with TKE values lower than 0.2 m² s⁻². It should be noted (Table 5) that the area with the lowest values of TKE in the resting corridors was increased by 51% in DRb3 compared to R3.

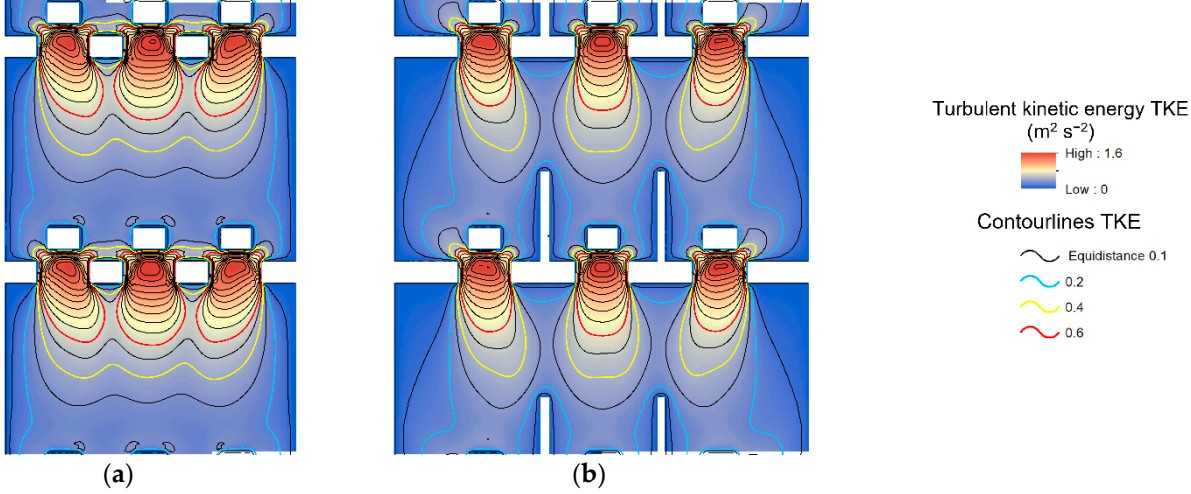

**Figure 13.** Turbulent kinetic energy in pool 3 and 4 for rectangular boulder layout (**a**), double rectangular boulder with a small barrier layout (**b**).

**Table 5.** Percentage of flow area occupied by each turbulent kinetic energy range and layout type: rectangular boulder layout (R3), double rectangular boulder (DR3), double rectangular boulder with a small barrier layout (DRb3) and double trapezoidal boulders (DT3). The percentage has been calculated respect of the pool area.

| TKE (m² s⁻²) | R3 | DR3 | DRb3 | DT3 |
|---|---|---|---|---|
| ≤0.2 | 17.65 | 29.87 | 36.09 | 26.96 |
| 0.2–0.4 | 42.56 | 44.28 | 38.98 | 43.63 |
| 0.4–0.6 | 17.20 | 11.99 | 12.09 | 16.00 |
| ≥0.6 | 22.59 | 13.86 | 12.84 | 13.40 |

## 4. Discussion

This section discusses the key question: Is there any alternative design which can improve the passage efficiency of uphill flow rock ramp?

As previously mentioned, the flow pattern in the pools is of great importance for guiding the fishes through the fishway. Both water depth and velocity primarily affect the swimming costs of fishes, although the turbulence level has also been shown to increase them considerably. Clay [44] has suggested that, when designing a fishway, it is necessary to locate the areas of low velocities (and also low turbulences) and assess how they can influence fish behaviour, since these are the hydraulic variables involved in passage efficiency.

In this paper, four different uphill flow ramp configurations were tested to show which design elements can positively impact the functionality of this type of ramp.

It is a prerequisite for all types of fishways that the depth must be sufficient for the fish to swim without difficulty, fully submerged, including the dorsal fin. The recommendation is for a depth of flow greater than, or equal to, two times the largest fish's body depth [4]. In the designs considered in this study, the median depth of the pools took the lowest value for DT3 (0.526 m) and the highest value for R3 (0.546 m). With this range of depths between models, transit was guaranteed for body depths from 0.26 m to 0.27 m. This small variation clearly showed that the design effect on depth was not relevant for fish passability.

Clay [44] has suggested that, when designing a fishway, it is necessary to locate the areas of low velocities and also low turbulences and assess the influence on the fish behaviour, since these hydraulic variables were critically involved in the passage efficiency. Regarding fishway design, the velocity in the gap (vmax) plays an important role, because it must present values lower than the sprint speed of the fish and with a passage time lower than the time the fish can maintain this velocity [45]. In all of the designs within this study, vmax was reduced with respect to the original R3 layout, getting the highest reduction (16%) in DT3. The absolute value of this variable should be interpreted in terms of the swimming capabilities of the target species. DT3 presented a median value of 1.52 m s$^{-1}$ (interquartile range: 1.15–1.81 m s$^{-1}$). Sanz-Ronda [45] studied two cyprinid species endemic to Spain and found total passability for lengths of 2 m and 4 m, with velocities of 2 m s$^{-1}$ and 1.5 m s$^{-1}$, respectively. Bearing in mind that the maximum length of vmax in this study was 1.3 m (length between diagonally opposite ends in the gap), this variable would not suppose any limitation for the passage of these species. These contributions allowed us to conclude that the shape of the boulders contributes to reducing the vmax in the gap.

The three different uphill flow ramp configurations, with regards to R3, including increased boulder size (DR3) and the incorporation of friction walls (DRb3), showed the influence of these geometrical aspects in the magnitude of turbulent kinetic energy. Turbulence and associated air entrainment is linked to the energy dissipation that occurs in the pools and is considered advantageous in the fish passage [23,46]. High turbulence may decrease swimming performance [47] and increase the energetic cost of swimming performance [46,48]. Fish have also exhibited preferences for low turbulence regions within fishways [49–51] and, in general, high turbulence levels seem to negatively affect fishway passage [52].

To evaluate this flow characteristic in one-dimensional hydraulic models, variables that measure it indirectly have been used: volumetric power dissipation (VPD), energy dissipation rate, or energy dissipation factor. These variables measure the power of the flow entering the pool, divided by the pool volume, W m$^{-3}$. The maximum recommended values vary according to the species or group of species: for cyprinids, a classical reference is 150 W m$^{-3}$ [53] and for salmonids, 200 W m$^{-3}$ [11], although other authors [54] raised this threshold to 240 W m$^{-3}$. However, the VPD assumes an average value of dissipated turbulence for an entire pool, omitting the distribution of turbulence [37], which can vary significantly across the pool.

TKE is a direct measure of the characteristics of the turbulence of the flow [55], quantifying the kinetic energy of the fluctuations of the components of the velocity vector. This energy is extracted from the mean flow due to shear between the mean and fluctuating velocities and gradients in the mean velocity field [56]. TKE has shown negative correlation

with the critical swimming speed of multiple fish species [47] and with the time needed for the fish to ascend a fishway [51].

Koziol [57], in a laboratory channel with a slope of 0.5 per thousand and relative depth 0.43 m, measured values up to 0.2 $m^2 s^{-2}$. Stone [58], in a gravel river (mean diameter 10.5 cm; flow 3.1 $m^3 s^{-1}$; slope 0.0012), measured values up to 0.05 $m^2 s^{-2}$ in the pool, 0.12 $m^2 s^{-2}$ in the run, and 0.33 $m^2 s^{-2}$ in the riffle. Puzdrowska [34], in laboratory model of ramps with aligned boulders, obtained the field of TKE values, with maximum values of 0.6 $m^2 s^{-2}$ and mean stream's flow area TKE does not exceed the value of 0.1 $m^2 s^{-2}$. Guiny [48] studied the passage of atlantic salmon with TKE 0.4–1.2 $m^2 s^{-2}$ in a pool with an orifice fishway. Silva [51] evaluated the passage of a cyprinid species, the Iberian barbel (*Luciobarbus bocagei* Steindachner 1864), in an experimental channel and found that the highest number of passages occurred at low TKE values (<0.05 $m^2 s^{-2}$, with high transit times variability). At $0.05 < TKE < 0.3$ $m^2 s^{-2}$ fewer fish passed, but transit time and variability were considerably reduced. The study recorded a passage with TKE = 0.36 $m^2 s^{-2}$ and one of the lowest transit times. Marrinier [59] has used TKE=0.05 $m^2 s^{-2}$ as a "low" threshold, referring to the results of Silva [51]. This value should not be interpreted as a threshold limit value but as preference value for passage.

Among the three designs, and in comparison to R3, DRb3 showed the greatest reduction in TKE when resting (median value TKE 0.13 $m^2 s^{-2}$) and intermediate corridors (median value TKE 0.28 $m^2 s^{-2}$) were considered together (Figure 12). Consequently, the flow area under this condition was 75%. This design also had the largest increase in surface area, with TKE < 0.2 $m^2 s^{-2}$ (36.09%, Table 5), and the largest surface area, with TKE $\leq$ 0.05 $m^2 s^{-2}$ (9.37%). These values were cited by Silva and Marrinier [51,59] as the preferred corridor. DRb3 was also the model with the lowest surface areas, with TKE above 0.6 (13%).

To summarise, this paper aimed to gain insights on potential alternative designs, able to outperform the passage efficiency of uphill flow rock ramp. Our results showed that:

(a)  in terms of uphill velocity and maximum velocity, the design with the greatest impact on the functionality of this type of ramp is the trapezoidal shaped double boulder design (DT3), as it maximized the uphill velocity in the resting corridors and, at the same time, has the maximum speed reduction in the row gaps. That is in accordance with Miranda et al.'s [17] statement: the obstacle shape is an important modifiable parameter for fishways designing.

(b)  in terms of the TKE, the most evident improvements were achieved in the resting corridor, with the double boulder layout (DR3), and with double boulder and friction wall in the intermediate corridors (DRb3).

Finally, these results provide integrated and tangible results that should be valuable to designing improved fish passes, aiming to maximize passability. Our results demonstrated that further investigation into the combined effect of the trapezoidal shape of the boulder, together with the friction barrier, is clearly needed due to their importance for increasing uphill velocities and reducing the TKE value, both in resting and intermediate corridors.

The comparison of the results obtained with the modelling of different designs provides useful information to be considered in the design of uphill flow ramps. However, it must be complemented with data that allow the evaluation of the fish passage performance (number of fish passing; passage time; number of attempts). Additional research, studying the biological response to changes in hydraulic variables, is needed to guarantee the improvement of future designs.

**Author Contributions:** Conceptualization, J.A.F.-Y. and C.M.S.-M.; methodology, J.A.F.-Y. and L.C.-D.; software, L.C.-D.; validation, J.A.F.-Y., C.M.S.-M. and L.C.-D.; formal analysis, L.C.-D.; investigation, J.A.F.-Y. and L.C.-D.; resources, L.C.-D.; data curation, L.C.-D.; writing—original draft preparation, L.C.-D.; writing—review and editing, J.A.F.-Y., C.M.S.-M. and L.C.-D.; supervision, J.A.F.-Y. and C.M.S.-M.; project administration, J.A.F.-Y. All authors have read and agreed to the published version of the manuscript.

**Funding:** This research received no external funding.

**Data Availability Statement:** Data are available upon reasonable request to the corresponding author.

**Acknowledgments:** We warmly thank Pepa Aroca for her technical support.

**Conflicts of Interest:** The authors declare no conflict of interest.

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
