# Peer review of "Uphill Flow Rock Ramps. How the Design Impacts Their Functionality"

_water, doi:10.3390/w14213492_

Round 1
Reviewer 1 Report
The present paper entitled " Uphill Flow Rock Ramp. How the design impacts their functionality" tried to discuss how the key hydraulic variables were impacted by different design element of uphill flow fishways with different configurations. The present paper can be considered to have content interesting for the journal. However, the paper needs some substantial revision for its approval.
Specific comments:
Abstract:
(1) Line 15. I did not find anything about theoretical calculations in present paper.
(2) Line 19. “…turbulent kinetic energy in resting corridors up to 33” should be “…turbulent kinetic energy in resting corridors up to 33%”.
Introduction:
(3) Lines 86-87 “Our aim here was to gain insights into the influence of several parameters in the fish passability and effectiveness of uphill flow ramps.” Actually, I did not find the studies on the influence of hydraulic variables on fish passablity in present paper, except water depth. I recommend to explain this information more clearly in “3. Results” or “4. Discussion”.
Materials and Methods:
(4) Line 109. “Therefor” may be replaced by “Therefore”
(5) Line 126. “…and placed in the correct distribution The uphill flow ramp design comprises three phases…” should be “… and placed in the correct distribution. The uphill flow ramp design comprises three phases…”
(6) Lines 136-137. The angle (α) and slop should be presented in Figure 3.
(7) Line 138. “… the dissipated power (400 W m-3).” Generally, the dissipated power should be less than 200 W m-3 in the pools of fishway (DVWK, 2002), why? Is it reasonable?
(8) Line 184. In Table 1, the parameters of barrier in design DRb3 may be added to differ DR3.
(9) Lines 193-198. The difference of the hydraulic characteristics between rock-weir fishways and vertical slot fishways is significant. Thus, it is not appropriate to discuss hydraulics together. Additionally, for rock-weir fishways, most studies used 3D model to simulate the hydraulics instead of 2D model. The author should further demonstrate the availability of 2D model in present study.
(10) “Model validation” should be added in “Numerical model”.
Results:
(11) Line 251. “The influence of pool, corridor and rock-row position” may be changed to “The influence of pool, corridor and rock-row position (R3).
(12) Lines 255-256. “Consequently, the velocity values were significantly higher.” The result "significantly higher" should be the results from the statistical test. Similarly, in Line 365, you also add the statistical test result.
(13) In Figure 7, 10, 12 and 14, the TKE value is much larger than 0.1m2 s-2, however, as discussed by Li et al. (2020) and Marriner et al. (2014), the value of turbulent kinetic energy should be lower than 0.05 m2/s2, and a negative correlation exists between fish transit time and turbulent kinetic energy levels.
Marriner, B. A., Baki, A. B. M., Zhu, D. Z., Thiem, J. D., Cooke, S. J., and Katopodis, C. (2014). Field and numerical assessment of turning pool hydraulics in a vertical slot fishway. Ecological Engineering, 63, 88-101. https://doi.org/10.1016/j.ecoleng.2013.12.010
Li, Y., Wang, X., Xuan, G., and Liang, D. (2020). Effect of parameters of pool geometry on flow characteristics in low slope vertical slot fishways. Journal of Hydraulic Research, 58(3), 395-407. https://doi.org/10.1080/00221686.2019.1581666
(14) Line 263. “…third quartile (75th percentile and outliers.” may be changed to “third quartile (75th percentile) and outliers.”.
(15) Lines 281-350. The meaning of the values in the figures, are they the average values of six pools or one pool or one corridor? Or are they the maximum value?
(16) In Figure 10, I recommend to add an explanation of “Fast corridors”, “Intermidiate corridors” and “Resting corridors”.
(17) Line 327. “…and 0.01 because de friction walls” should be “…and 0.01 because of friction walls”.
Discussion:
(18) Scientific discussion should be performed more adequately. In my opinion, the authors should add the information of the influence on the fish passability and effectiveness of uphill flow ramps.
References:
(19) The page range is missing for most of literatures.
Reviewer 2 Report
General comment:
The authors present the results of numerical simulations of hydrodynamics of different fishway setups. While interesting and valuable, the paper needs significant improvements and re-arrangements before publishing.
Specific major issues:
(91) There are a few possibilities of defining turbulent kinetic energy, especially in a 2D approach. A formula is needed, especially because this variable is one of the key ones in this study (229+).
(107-110) If the first layout (R3) reproduced a real life structure, than it should be used to calibrate and/or validate the model. If it is not done, than even “relative data” as the authors write in (109) are doubtful. Comparison of model results for the R3 setup with measurements made for the structure on the Manzanares River (at least on a qualitative level if quantitative is impossible) is necessary.
(118) Ambiguous descriptions; especially no explanation what “velocity y-axis” stands for. Needs reworking.
(207) There are a couple of forms “shallow water equations” can have. Despite referencing the model-related papers, presenting the governing equations is welcome here.
(215) In 2D models the Manning coefficient can be significantly higher than expected because it often serves also as a damper of numerical instabilities. A reasoning behind the chosen value, preferably with the model sensitivity analysis with respect to this parameter, is needed.
(217) The meaning of “validated” in this context is obscure. Isn’t it just stopping the simulation when it gets convergence here?
(250-353) I am not sure why the real modeling results (like Fig 15, line 399) are only very briefly presented at the end of the “discussion” chapter, while the statistical analysis and discussion of the results is presented in the “results” chapter. I strongly recommend presenting much more of the straightforward modelling results (with appropriate graphs), compacting the statistical part (especially the figures in the current 3.2 chapter which are sub-optimal now), and moving it to the discussion section. Moreover, while it is fine to write that a certain statistical test gave such-and-such result, it should be followed by a few words of conclusion on “what does it mean in the context of the matter our study”.
Specific minor issues:
(18, 104) Starting the enumeration with ‘i)’ in the middle of the text makes the reader unsure what is going on: has the opening parenthesis got lost somewhere? What ‘that i’ stands for? It interrupts the flow of the information. I recommend putting a colon after “that” and using “1), 2) 3)” instead of “i), ii)” etc.
(67,100) The web link would fit better in the references section not in main text.
(66-79) Figure 1 would fit better somewhere here than in (118).
(102-106) Appropriate figures should be called out here.
(106-117) Language proofreading is needed for this fragment.
(237 – in the table) vy should be references as a component not a vector.
(388-396) This fragment deserves some expanding as a conclusions section.
Reviewer 3 Report
First of all, I think this is a very good high level paper, the author puts forward the boulders Zig-Zag arrangement for nature-like fishways has important value, I completely agree with the author for the flow structure parameter selection and corresponding calculation methods, I look forward to the author can show the related research results on actual target migration effect.
In Table 1, DT5 should be DT3. In this layout type, the size of the boulders on both sides should be Db*=1.25m & Db=1.35m,Please check and mark in the table.
Round 2
Reviewer 1 Report
The manuscript has improved with respect to the previous version, and corrected most of the previously mentioned deficiencies.
A few more corrections/explanations are suggested:
(1) “y” represents water depth in L297, however, it is “y” axis in L293-294, I recommend to change it.
(2) L358 “they are below the detection limit of the variable in field (2 cm).” I don’t understand clearly how the 2cm limit is determined here?
(3) In the manuscript, “Vy”, “vy”, “vy” … should be unified if they all represented the same variables. The same problem for “vmax”, “vmax”, …
(4) In discussion, L531-547 should replaced in the part of numerical model, they are more methodological explanations than discussions.
(5) L574 “vma” should be changed to “vmax”
(6) The graphical quality of figures in present manuscript should be improve further, the resolution is low especially the text in the figure.
Reviewer 2 Report
In my opinion all the issues pointed out in the first iteration have been sucesfully resolved. I'd recommend showing velocity fields figures before any statistical analysis results but it is not necessary.
